# Effects of Weight Loss on Endothelium and Vascular Homeostasis: Impact on Cardiovascular Risk

**DOI:** 10.3390/biomedicines13020381

**Published:** 2025-02-06

**Authors:** Margherita Tiezzi, Francesco Vieceli Dalla Sega, Paolo Gentileschi, Michela Campanelli, Domenico Benavoli, Elena Tremoli

**Affiliations:** 1Dipartimento Cardiovascolare, Maria Cecilia Hospital GVM Care and Research, 48033 Cotignola, Italy; etremoli@gvmnet.it; 2Laboratorio di Ricerca Traslazionale, Maria Cecilia Hospital GVM Care and Research, 48033 Cotignola, Italy; fvieceli@gvmnet.it; 3Dipartimento di Chirurgia Bariatrica e Metabolica, Maria Cecilia Hospital GVM Care and Research, 48033 Cotignola, Italy; gentileschi.paolo@gmail.com (P.G.); michelacampanelli@live.it (M.C.); dobenavoli@yahoo.com (D.B.); 4Dipartimento di Scienze Chirurgiche, Università di Roma Tor Vergata, 00133 Roma, Italy

**Keywords:** weight loss, endothelial function, bariatric surgery, biomarkers

## Abstract

Available knowledge shows that obesity is associated with an impaired endothelial function and an increase in cardiovascular risk, but the mechanisms of this association are not yet fully understood. Adipose tissue dysfunction, adipocytokines production, along with systemic inflammation and associated comorbidities (e.g., diabetes and hypertension), are regarded as the primary physiological and pathological factors. Various strategies are now available for the control of excess body weight. Dietary regimens alone, or in association with bariatric surgery when indicated, are now widely used. Of particular interest is the understanding of the effect of these interventions on endothelial homeostasis in relation to cardiovascular health. Substantial weight loss resulting from both diet and bariatric surgery decreases circulating biomarkers and improves endothelial function. Extensive clinical trials and meta-analyses show that bariatric surgery (particularly gastric bypass) has more substantial and long-lasting effect on weight loss and glucose regulation, as well as on distinct circulating biomarkers of cardiovascular risk. This review summarizes the current understanding of the distinct effects of diet-induced and surgery-induced weight loss on endothelial function, focusing on the key mechanisms involved in these effects.

## 1. Introduction

Obesity is a systemic disease characterized by an excessive accumulation of body fat, usually identified by a body mass index (BMI) above 30 kg/m^2^. Additional indicators, like waist circumference, waist–hip ratio, and the percentage of body fat, are used to better define visceral adiposity and improve CV risk prediction [1,2,3]. Recently, the prevalence of obesity has experienced an unstoppable growth, with 41% of the United States (U.S.) population being overweight (BMI > 25 kg/m^2^) and around 16% being obese (https://www.tfah.org/report-details/state-of-obesity-2022/, accessed on 27 January 2025). This is expected to increase, especially in women and in low economic status populations [4]. Obesity is associated with high morbidity and mortality, and affected patients may present diabetes mellitus (DM), arterial hypertension, and atherosclerosis, with an enormous impact on the cardiovascular (CV) prognosis. The clustering of obesity (especially visceral), with CV risk factors such as insulin-resistance, dyslipidaemia, and hypertension, defines the metabolic syndrome, which further contributes to the CV burden. The pathogenesis of obesity is multifactorial, involving dietary, socioeconomic, genetic, and metabolic factors [5]. Adipose tissue has different structural and functional characteristics: white adipose tissue (WAT), mostly visceral, contains a single lipid droplet per adipocyte and is associated with an increased vasoconstriction and an increased CV risk, whereas brown adipose tissue (BAT) contains multiple lipid droplets and a large number of mitochondria, bearing a thermogenic, anti-inflammatory function; ultimately, beige adipose tissue represents an intermediate entity, with cardioprotective functions [6] (Figure 1). In obesity, the accumulation of visceral WAT has proinflammatory activity, due to the enhanced production of pro-inflammatory cytokines (adipocytokines), which contribute to the local recruitment of activated inflammatory macrophages [7]. Endothelial dysfunction, defined as the inability of the endothelium to regulate the main vascular functions (vasodilation, coagulation, permeability), is considered a pathological event that precedes atherosclerosis. In obesity, systemic inflammation, adipose tissue dysfunction, free radical production, the concomitant presence of insulin resistance, and elevated levels of oxidized LDL all contribute to an impaired endothelial homeostasis, a critical event that occurs well before overt CV disease [8].

The increasing interest in studying endothelial pathology reflects the need for clinicians to improve CV prevention and prognosis. Endothelial homeostasis can be assessed by different methods that represent different “windows” to look at systemic alterations. These techniques can be summarized in (1) functional assessments, aimed at measuring endothelial vasodilator response, permeability, and anti-thrombotic activity in vivo, i.e., endothelial function [9], (2) circulating biomarkers (soluble adhesion molecules, nitric oxide (NO) metabolites, and circulating endothelial cells), which provide an insight in endothelial activation and turnover and can give a hint of possible damage mechanisms [10,11], and (3) in vitro studies which allow us to explore the mechanisms of endothelial injury/dysfunction and to perform a “liquid biopsy” of the patient’s endothelium (primary cultures) [12]. In this context also vascular remodelling—defined by the presence of structural changes of the entire vascular wall—is of certain interest, even if it is not a direct measure of endothelial behaviour but rather a consequence of the structural modifications involving the intima and media layers of the vessels’ walls. 

Weight reduction approaches, based on lifestyle advice or surgical treatments, have shown an undeniable beneficial role in metabolic status and systemic inflammation, and a role in the improvement of vascular homeostasis and the CV outcome [13]. Dietary regimens are primarily based on lowering food consumption and adjusting nutrient ratios. Bariatric surgery (BS) is a highly effective weight-loss method, primarily achieved through the reduction in stomach capacity or the creation of a digestive bypass. While lifestyle interventions such as diet have been proven to reduce CV events and death, especially when reaching a threshold of a 10% weight reduction, BS has been reported to have a more important impact on all-cause mortality, CV mortality, and the incidence of heart failure, myocardial infarction, and stroke, as compared to standard care [14,15].

This review aims to summarise the available knowledge on the effects of weight loss on the endothelial function and homeostasis, considering various methods for achieving a significant weight reduction, and attempts to elucidate the apparent greater influence of BS on the CV system. While recognizing the impact of approved drugs to treat obesity, this review will only focus on the non-pharmacological treatments.

## 2. Diet-Induced Weight Loss and Vascular Endothelium 

Diet and/or dietary composition have a significant impact on vascular health, and the Mediterranean diet in particular has been shown to possess beneficial effects on the endothelial function [16] and has been proven to prevent CV events in high-risk populations [17].

Different dietary regimens have been proposed to achieve a significant weight reduction: a very-low-caloric diet, a low-fat diet, a low-carbohydrates diet, a low-fat high-carbohydrates (CHO) diet, to cite a few. The evaluation of the effects of the different diets is complex. The most consistent data indicate that the Mediterranean diet is associated with CV protection. It is more difficult to discern the endothelial impact of low-carbohydrates diets due to the variable use of animal proteins that can modify the CV effect [16,18,19]. For the purpose of this review, we will consider all the dietary regimens together, provided that an effective weight loss was achieved at the end of the observation period.

Diet-induced weight loss (DIWL) is normally estimated at around 5–17%, depending on the study design, populations, and food restriction approaches [20,21]. Available evidence shows that DIWL is associated with a broad spectrum of beneficial metabolic changes, including oxidative stress reduction, increased mitochondrial energy efficiency, and better control of CV risk factors [22]. Similarly, DIWL leads to a significant improvement in endothelial homeostasis [23]. 

It is difficult to measure the impact of different dietary regimens on CV health because diets are often associated with physical activity, which per se may reduce body weight and influence endothelial homeostasis through different mechanisms, thus promoting positive CV effects [24,25].

### 2.1. Endothelial Function in DIWL

Studies report that after DIWL, an improvement of endothelial functional capacity is observed. Flow-mediated dilation (FMD) of the brachial artery is the most frequent and most accessible method to assess the endothelial function, and its decline is correlated to an increased CV risk [26]. Thus, FMD reflects endothelium-dependent vasodilation, e.g., the capability of endothelial cells (ECs) to promote vasodilation (via nitric oxide, NO) in response to hyperaemic stimulus. Raitakari et al. showed a 60% improvement in FMD, compared to baseline, after 6 weeks of a very-low-caloric diet, regardless of sex, smoking and presence of systemic hypertension. These changes were associated with a concomitant decrease in plasma glucose levels, as well as an increase in adiponectin levels [21]. In line with these findings, a large metanalysis by Joris et al., carried out on patients treated with diet alone and/or BS, showed a significant increase in FMD after weight loss. However, heterogeneity in weight reduction regimens and the small proportion of studies including BS limit the significance of the results [27]. In addition, in this metanalysis, healthy individuals and patients with CV comorbidities or diabetes were included, and several trials were sex-specific, including only men or women. In a study where FMD was evaluated pre- and post-DIWL in a population of healthy men following a 6-week caloric restriction program, in comparison to a “no-weight loss” group, no effect of FMD was observed [28]. Thus, the heterogeneity of the studies, the dietary measures, and populations likely plays a role in the conflicting results observed in FMD.

### 2.2. Circulating Biomarkers in DIWL

The effect of DIWL on the levels in plasma of biomarkers potentially reflecting the behaviour of vascular endothelium has been extensively studied. In particular, the effect of different dietary regimens was tested in a metanalysis published by Mathur et al. [29]. Overall, a consistent decrease in circulating intercellular adhesion molecule 1 (ICAM1), vascular cell adhesion molecule 1 (VCAM1), E-selectin, and vascular endothelial growth factor (VEGF) were observed, with a concomitant increase in nitrite/nitrate (NOx) after a low-caloric diet during a period ranging from 3 to 52 weeks. ICAM1, VCAM1, and E-selectin, expressed on activated ECs, are useful markers of endothelial dysfunction and activation [30]. E-selectin is an endothelial cell surface molecule involved in neutrophil adhesion; it is upregulated on the endothelial cell surface in response to inflammatory stimuli, such as interleukin 1 and tumour necrosis factor alpha (TNF), and altered shear stress, in a tight cross-talk with vascular smooth muscle cells (VSMCs) [31,32]. Thus, E-selectin is a marker of activated endothelial cells [33]. VEGF belongs to a family of powerful angiogenic factors promoting neovascularization, vascular permeability and EC proliferation [34]. Finally, NOx are NO metabolites, and their increased levels may reflect increased NO production by vascular endothelium and bioavailability. 

DIWL has been reported to markedly reduce systemic inflammation as a result of weight loss and of the type of diet adopted [22,35]. A randomized controlled trial, including 93 obese subjects undergoing a very low-energy diet, showed that at least a 10% of weight loss is required to significantly reduce C-reactive protein (CRP) levels in plasma [36]. Further research, carried out on subjects undergoing either a 6-week low-caloric diet or a 12-week very low-fat diet, confirmed the significant reduction in systemic inflammation after weight loss [21,37].

Thus, the effects of DIWL on endothelial behaviour include both the normalization of vascular function and of circulating biomarkers, which are parallel to the improvement of systemic inflammation, in turn contributing to reset vascular homeostasis. The heterogeneity of study designs, the different time points analysed, and different populations are all factors that should be taken into account in the interpretation of these data.

Figure 2 summarizes the available evidence on DIWL and CV health. 

Interestingly, studies on DIWL showed that FMD improvements do not correlate with weight reduction but rather with glucose levels [21,38]. Also, Mathur et al. showed no correlation between the levels of endothelial circulating biomarkers and the weight loss percentage [29]. The same is true for BS studies, where improvements in FMD were shown to be independent of body weight reduction [38,39].

These somehow conflicting data suggest that additional factors, as, for instance, improved glycaemic control, might represent the main factors in CV improvements after weight loss. Also, waist circumference, rather than weight loss or BMI, represents a more reliable parameter reflecting CV health, but unfortunately it was not used in all the studies. The role of glucose metabolism on vascular health is discussed in a separate paragraph below.

## 3. Surgery-Induced Weight Loss and Endothelial Function 

BS includes all the surgical techniques aimed at reducing body weight. According to the guidelines, BS can be applied to patients with BMI ≥ 35 kg/m^2^, or >30 kg/m^2^ in the presence of associated comorbidities [40,41]. There are various mechanisms of action in BS procedures: restrictive, malabsorptive, and hybrid (hormonal neurotransmitter-mediated). All procedures act differently on gastro-intestinal physiology, with concomitant effects, which can be both restrictive and hormone-related. The mainly restrictive operations (sleeve gastrectomy, gastric banding and endoscopic gastroplasty) aim at reducing the gastric volume while maintaining anatomical intestinal transit, whereas the malabsorptive techniques (including the gastric bypass and all its variants) lead to the exclusion of part of the gastrointestinal tract, with altered micronutrient absorption and an increased risk of malnutrition (Figure 3). 

Surgery-induced weight loss (SIWL) is estimated to result in a reduction of 50 to 77% of excess body weight (EBW) [42,43,44]. This drastic weight reduction is normally obtained in the first 6 to 12 months from surgery, with significant differences between surgical techniques. Regardless of the technique used, BS has been proven to consistently reduce CV events and mortality, as compared to lifestyle interventions, especially in the presence of DM [13,45].

### 3.1. Endothelial Function in SIWL

Endothelial functional capacity is improved after SIWL. A metanalysis from Lupoli et al., including eight studies mainly using the Roux-en-Y gastric bypass (RYGB), showed that, after 3 to 24 months from surgery, RYGB was associated with an overall improvement in FMD, especially in the first year after surgery, with BMI, body weight reductions, male gender, dyslipidaemia, and impaired fasting glucose being the main predictors of FMD improvement [46]. It should be considered, however, that this metanalysis included a significant number of studies involving diabetic patients, known to have altered endothelial function not related to obesity. In a recent metanalysis, which involved mainly non-diabetic patients who underwent RYGB, the overall FMD improved from 6 to 12 months from surgery, but no association was found between the BMI changes and the FMD of the brachial artery [39]. Instead, an overall improvement in the endothelial functional capacity after BS was found in the coronary bed [47,48]. 

Few studies compared the effect of DIWL with that of SIWL on endothelial functional capacity. Surgical treatment with RYGB was compared to diet interventions, in a small trial, with a 3-month follow-up. In this study, both SIWL and DIWL significantly increased FMD, with the gastric bypass surgery being more effective. In both cases, no correlation was found between FMD changes and body weight reductions, while a strong negative correlation was reported with fasting glucose levels [38]. The same team evaluated DIWL and SIWL at a 12-month follow-up and found a significant increase in FMD in patients achieving at least 10% of weight loss, regardless of the technique used [49].

### 3.2. Circulating Biomarkers in SIWL

SIWL is reported to be associated with the decrease in several circulating biomarkers involved in vascular homeostasis. A metanalysis of 771 studies included patients undergoing BS, with follow-ups from 15 days to 84 months post-surgery (restrictive and malabsorptive techniques), and indicated a significant reduction in ICAM-1 and PAI-1 after BS [50]. These data were confirmed by another metanalysis carried in men and women undergoing BS (mostly RYGB). Subgroup analyses did not find differences between different sex, comorbidities or surgical techniques [51]. It should be noted, however, that most of the above-mentioned data refer to metanalyses that include different study designs, surgical techniques, and follow-up periods, all factors that should be taken into account in our evaluation.

Systemic inflammation is a major determinant of cardiovascular health, and increased levels of inflammatory cytokines are often found in the blood of obese individuals [52]. The modifications in inflammatory markers after SIWL may likely play a role in CV risk. In a prospective study carried out on severely obese patients (40% diabetic, 60% female), a reduction in MCP-1, IL-6, CRP, ferritin, and PAI-1 was observed 1 year after the sleeve gastrectomy [53]. In a metanalysis performed by Rao on 48 prospective studies (mainly RYGB), the decrease in CRP and IL-6 levels were confirmed during follow-ups after 1 to 48 months [54]. These findings have been confirmed by a subsequent metanalysis, in which a significant decrease in TNF after BS was reported [50,55]. RYGB, in particular, seems to induce a more pronounced improvement in systemic inflammation and metabolic profile, which is also maintained in the long term and regardless of additional behavioural measures [56]. Figure 3 summarizes the available evidence concerning SIWL and CV health.

In summary, the available knowledge suggests a beneficial role of both DIWL and SIWL on circulating endothelial markers and endothelial homeostasis, despite the heterogeneity of patient characteristics, study design, and therapeutical measures. SIWL, compared to DIWL, induces earlier and more drastic vascular and metabolic modifications, explaining the increased CV protection observed. However, it is essential to perform further studies to assess the relative impact of diet or exercise on vascular endothelium, according to the different BS treatments used, taking into account the fact that RYGB is the most represented surgical technique. Dietary measures and surgical interventions are likely to result in distinct hormonal and metabolic alterations, which may in turn contribute to cardiovascular effects.

## 4. Mediators Linking Weight Loss and Vascular Endothelium

As discussed above, weight loss and obesity are associated with intricate hormonal and metabolic shifts that extend far beyond the digestive system. Research has found that multiple mediators are disrupted in obese individuals, and studies are focused on discovering these changes in obesity in relation to the weight loss achieved. Research in this area is still ongoing, and we reviewed the available evidence concerning the primary mediators that play a role in endothelial health before and after DIWL and SIWL.

### 4.1. GLP-1

Glucagon-like peptide 1 (GLP-1) is a hormone secreted by the gastrointestinal tract and the central nervous system (CNS) in response to food ingestion [57]. Its role consists in promoting satiety (inhibiting gastric emptying) and improving glycaemic control by stimulating insulin secretion and inhibiting glucagon production [58] (Figure 4). In normal-weight individuals, GLP-1 levels are very low after overnight fasting, increase rapidly with food intake, and do not return to the fasting levels during the day [59]. In obese patients, lower levels of GLP-1 are observed, especially in the post-prandial period [60]. The mechanisms behind decreased GLP-1 production remain unclear, but this factor might contribute to an altered “satiety” response in obese individuals. The wide distribution of the GLP-1 receptor (GLP-1R) over different organs accounts for the pleiotropic actions of this hormone, notably on the CV system. Indeed, GLP-1 action encompasses cardiomyocyte protection and improved endothelial function, promoting cell survival, reducing oxidative stress, and increasing NO production [61]. Long term dietary programs have been proven to significantly increase GLP-1 levels in obese subjects [62,63], and a 2-year dietary program conducted on obese women confirmed the significant post-prandial increase in GLP-1 associated with the amount of weight loss and influenced by the dietary regimen [64]. However, these findings have not been confirmed in other studies [65,66,67]. Finally, Aukan et al. very recently showed a significant increase in post-prandial GLP-1 after sleeve gastrectomy and RYGB, but not after diet alone [68].

Generally speaking, an overall increase in post-prandial GLP-1 levels is reported after BS [69]. RYGB and sleeve gastrectomy showed the most consistent results, with a more rapid and pronounced modification of GLP-1 levels after surgery, up to 1 year after the intervention, as compared to diet alone [68]. Also, after a biliopancreatic diversion/duodenal switch (BPD/DS, mixed restrictive/malabsorptive technique), increased post-prandial GLP-1 levels were observed up to 2 years from surgery, which positively correlate with the percentage of weight loss [70]. Interestingly, gastric bypass has been shown to induce a significant increase in postprandial GLP-1, but not gastric banding [71]. Other studies on RYGB patients showed a lower GLP-1 post-prandial increase in patients with poor weight loss, compared to patients reaching the goal of a robust weight loss (usually >30 to 60% of total body weight) [72,73].

Given the role of GLP-1 on the CV system, it is reasonable to speculate that BS-induced GLP-1 increase contributes to an improved vascular health. Interestingly, studies in animal models have suggested that GLP-1 changes might directly have a role in the cardioprotective effects occurring after BS. GLP-1 levels rapidly increased in obese rats after RYGB, which was directly associated with improved endothelial function and independent of weight loss [74]. Additionally, a role of GLP-1 in the stimulation of WAT browning, with an improvement in lipolysis, fatty acid, and glucose metabolism, has been suggested, based on experimental studies on animals [75,76]. Data acquired from humans confirm this observation, showing positive changes in BAT metabolic activity in healthy subjects treated with GLP-1R agonists [77], suggesting that at least a part of GLP-1’s positive CV effects are mediated by either the WAT browning or by an increased BAT function.

### 4.2. Ghrelin

Ghrelin is an orexigenic hormone secreted by the gastric fundus and the duodenum in response to a negative energy balance (“empty stomach”) [78,79]. Its “classical” action is appetite stimulation via the interaction with neurons in the hypothalamic arcuate nucleus, which produce anabolic mediators, such as the agouti-related protein (AgRP) [80]. Circulating levels of ghrelin correlate inversely with the body mass index and have been found to be reduced in obesity [81]. Ghrelin is produced mainly by the gastric system (<60–70%) but also by other organs, including kidneys, lungs and hypothalamus. Ghrelin receptors (GHS-R) are located not only in the hypothalamus, but also in the different areas of the CV system [82]. Circulating ghrelin exists in two forms: unacylated (UAG) or acylated (AG), but the research on the role of each isoform in the different areas of the cardiovascular system is still ongoing. In healthy subjects, acylated ghrelin improves cardiac functional parameters, but not endothelial function [83,84]. Similarly, in patients with heart failure acylated ghrelin significantly improves cardiac function [85]. On the other hand, unacylated ghrelin significantly ameliorates endothelial-dependent vasodilation, increasing NO availability in metabolic syndrome patients [86], and improves endothelial cells survival in vitro [87]. Also, ghrelin was shown to inhibit proinflammatory responses and directly promote NO production within endothelial cells in vitro. Unfortunately, in this study the type of ghrelin isoform was not specified [88].

Ghrelin is also known to have a role in glucose metabolism, and studies in human and animal models have shown that it alters glucose tolerance and increases insulin levels. Interestingly, it has been shown that acylated ghrelin worsens glucose tolerance [81], whereas, in a co-administration with the unacylated form, it significantly improves insulin sensitivity [89] (see Figure 5 for a schematic representation of the CV effects of ghrelin).

Overall, one can hypothesise that the unacylated form of ghrelin has vasculo-protective effects, whereas its acylated form elicits cardioprotective actions directly, negatively impacting glucose metabolism [90]. It should be noted that patients may differ according to their capacity to produce the unacylated form of ghrelin, or alternatively, they may differ according to their capacity to produce the enzyme ghrelin O-acyl-transferase (GOAT), with a consequent unbalance of ghrelin isoforms. Further studies will allow to differentiate the role of these isoforms on the CV system in healthy and diseased patients.

DIWL induces a compensatory increase in ghrelin levels, leading to a frequent sensation of hunger and, thus, a higher likelihood of weight regain. On the contrary, BS (especially RYGB) is associated with suppressed ghrelin levels and a modification in eating behaviour (loss of appetite, consumption of low-caloric foods), probably secondary to gastric fundus removal [20]. The comparison between the effects of different BS techniques (sleeve gastrectomy, RYGB) on ghrelin levels has provided conflicting results, showing either an increase in post-prandial levels in both RYGB and sleeve gastrectomy, no effects, or an isolated increase after sleeve gastrectomy alone [91,92,93]. The fact that DIWL and SIWL are associated with opposite modifications in ghrelin levels while maintaining a beneficial effect on the CV system is interesting, and hints at the different adaptive mechanisms following these weight loss techniques. The earlier and more drastic improvement in glucose metabolism observed after BS is probably at least partly mediated by the suppression of ghrelin levels after surgery. Also, the differential outcome of the ghrelin infusion in healthy or metabolic syndrome patients may indicate the presence of a threshold effect or a variable GOAT enzymatic activity.

### 4.3. Adiponectin 

Adiponectin is a hormone mainly produced by adipocytes, with a well-recognised role in inflammation and glucose sensitivity [94]. It modulates and inhibits all the atherogenesis steps, starting from endothelial injury to LDL uptake and macrophage infiltration [95]. In physiological conditions, adiponectin inhibits the adhesion molecules expression and apoptosis and increases NO bioavailability in endothelial cells [96,97]. It also reduces macrophage-to-foam cell transformation, VSMCs proliferation and migration, and blocks nuclear factor kappa B (NF-kB) activation in ECs, modulating the inflammatory response within the vessels’ walls [98]. Adiponectin levels are decreased in obesity, diabetes, metabolic syndrome, coronary artery disease, and systemic hypertension [99,100]. A low level of adiponectin in plasma due to missense mutations of the adiponectin gene contributes to higher incidence of CV diseases and diabetes [101]. Moreover, its plasmatic concentrations can be restored by using Renin-Angiotensin System (RAS) blockers or by anti-hypertensive combination therapy [102]. Evidence from both human and animal models is summarized in Figure 6. 

The amount of weight loss achieved and the type of diet after DIWL have been shown to influence adiponectin levels. Ratliff et al. reported a 21% increase in adiponectin levels in overweight men after a carbohydrate-restricted diet for 12 weeks, even with a mild reduction in body weight (~6%) [35]. In a study carried out in obese men undergoing a very-low-caloric diet, it was observed that at least 10% of weight loss is required for a short- or long-term increase in adiponectin levels [36]. A similar study confirmed the significant increase in adiponectin levels after a 14-week hypocaloric diet, an effect related to the changes in the lipid profile. In that study, an inverse correlation between adiponectin concentrations and HDL catabolism was reported [103]. Similar data were found in a study carried out in obese women undergoing a low-caloric diet [104].

Adiponectin levels globally increase after BS [50,105,106]. Studies on sleeve gastrectomy (the most common restrictive technique) indicate a significant increase in adiponectin concentration after weight loss, even if to a lesser extent when compared to other surgical approaches [106,107]. Interestingly, studies involving mixed restrictive/malabsorptive BS (such as biliopancreatic diversion with duodenal switch) showed conflicting results [108,109].

The changes observed in adiponectin concentrations support a stronger effect of SIWL, compared to DIWL. The stimulus leading to increased adiponectin secretion from adipose tissue remains, however, a matter of debate. The reduction in dysfunctional adipocytes and the restoration of AT physiological secretory activity following weight loss might possibly restore physiological adiponectin production as well. These changes might also explain the wider cardioprotective action of BS, which indeed proved to be more beneficial than DIWL in reducing CV events [14].

### 4.4. Leptin 

Leptin in an adipocyte-derived hormone involved in energy balance and bearing an anorexigenic function. It is mainly secreted by adipose tissue (especially WAT) under different stimuli, such as food intake, increased insulin concentrations and production of pancreatic peptides [110]. Similarly to other previously described mediators, leptin acts via Ob-R receptors that exist in six isoforms and account for its pleiotropic action. Increased levels of leptin are independently associated with insulin resistance, cardiovascular disease, CV events, and the extension of coronary artery disease [111]. Moreover, its structural analogy with C-reactive protein suggests a possible role of leptin in modulating systemic inflammation, providing a rationale for its CV effects. 

Obese patients display higher levels of circulating leptin, secondary to the so-called “leptin resistance”. Such condition is characterised by the inability of leptin to reach the target organs/cells, leading to the loss of efficacy of both endogenous and exogenous leptin on target organs. Recent evidence from animal models shows that leptin-signalling impairment leads to an increased vascular neointima formation, thus altering endothelial homeostasis [112]. In humans, a leptin resistance adversely affects various tissues including the vasculature, but it is not clear whether this is dependent on a resistance to leptin or on an excess of leptin action. Available data indicate that there is a threshold effect related to leptin levels, implying that any variation from typical levels might lead to compromised CV health: according to studies mainly conducted on animal models, elevated leptin levels cause cardiac hypertrophy; however, these changes are reversible once normal leptin levels are restored [113]. Also, data from preclinical studies indicate that adipocyte-derived leptin induces endothelial dysfunction, whereas endothelium-derived leptin has protective effects on it, with significant differences in males and females [114] (Figure 7). 

DIWL (with or without exercise) induces a reduction in leptin levels [115]. This reduction has been shown to parallel the improvement in the obesity-associated procoagulant phenotype in a study on obese women undergoing very-low-caloric diet [116]. However, circulating leptin levels are deeply affected by dietary composition [117]. Leptin levels are significantly reduced after malabsorptive BS, especially within the first 12 months of follow ups [50,118]. Interestingly, this reduction correlates with the degree of improvement in carotid intima-media thickness at 2 months post-surgery [119]. The same trend was found after restrictive BS, such as vertical banded gastroplasty, where leptin was shown to progressively decrease up to 48 months after surgery [120]. A prospective trial carried out on 148 patients undergoing sleeve gastrectomy revealed a non-significant decrease in leptin levels at 12 months; however, when considering the leptin/adiponectin ratio, the reduction was significant and mainly driven by an adiponectin increase [121]. These partially conflicting data probably result from the fact that, in humans, the levels of leptin may vary according to the population studied. Interestingly, females were shown to have increased leptin levels in plasma, compared to males.

Interestingly, studies on animal models showed that a decrease in circulating leptin is required to promote the positive effects of GLP-1, such as weight reduction, increased insulin signalling, and systemic inflammation [122].

### 4.5. Glucose 

An improved glucose metabolism, after either DIWL or SIWL, has positive effects on CV health through the prevention of pathological arterial remodelling and atherosclerosis. Insulin stimulates the PI3K/Akt pathway, directly promoting the NO synthase activation, with a consequent vasodilation, due to the increased glucose uptake [123]. In parallel, it induces VEGF and ET-1, promoting neoangiogenesis and vasoconstriction. An impaired glucose tolerance directly affects endothelial cells, altering the balance of vasodilator synthesis and/or release, with a reduction in NO levels and an even more significant decrease in ET-1 in experimental animals, through mechanisms that have not been as yet fully elucidated [124]

The beneficial effect of DIWL on glucose metabolism is not a consistent finding and this variability probably depends on study designs and follow up duration [20,125].

Interestingly, data concerning the effect of SIWL on glucose or insulin levels are consistent, suggesting that the surgical procedure has a beneficial role. SIWL (especially RYGB) is associated with a pronounced and early improvement in insulin sensitivity, fasting glucose and insulin secretion, together with an increase in glucose transporters, with frequent cases of type II DM remission after surgery [126]. The comparison between surgical techniques confirmed the earlier improvement (1 week after surgery) in glycaemic control after RYGB, compared to sleeve gastrectomy. However, both restrictive and malabsorptive techniques significantly ameliorate glucose metabolism during both early and late follow up [50,91].

## 5. Conclusions and Gaps in Evidence

The available knowledge strongly supports the benefit of weight loss on the vascular endothelium in overweight and obese adults, treated with a diet alone, or in combination with BS, when appropriate. Direct and indirect signs of endothelial dysfunction, which occur in obesity and overweight, are both modified by DIWL and SIWL, and data indicate that inflammation and glucose levels are one of the main drivers that alter endothelial function. Available data also suggest that weight-independent mechanisms sustain improved endothelial function after weight loss. The faster improvement in glucose metabolism occurring in patients after BS due to the weight loss and partially due to the suppressed ghrelin levels, and the reduction in systemic inflammation, may collectively account for the amelioration of the endothelial function and the reduction in CV outcomes following BS. We should also consider that BS patients undergo significant modifications in gastric volume and appetite, all factors that further contribute to the long-term maintenance of the achieved body weight, limiting the rebound effect observed in DIWL. 

However, it should be noted that the gaps in the evidence concerning weight loss approaches and endothelial function protection are still multiple: (1) DIWL studies are non-standardised, include different dietary regimens (which may differentially affect endothelial function), and often are associated with exercise; (2) only a few studies analyse the results while differentiating between overweight and obesity, as well as hormonal and non-hormonal obesity; (3) DIWL and SIWL studies differ on follow-up lengths and are often gender-specific, including a large proportion of women, especially in BS. Finally, (4) the role of possible nutrient malabsorption on endothelial function assessment after BS has not been systematically investigated and represents a possible confounding element, especially in studies analysing early time points. 

Further research is needed to unveil the metabolic mechanisms involved in the restoration of endothelial function after weight loss, according to the different diets or surgical approaches. Also, the role of sex and genetic predisposition in obesity prevalence and the metabolic and hormonal responses to weight loss must also be taken into account [127,128].

Thus, on the basis of the available knowledge, we can conclude that the beneficial effects of BS do not rely solely on food restriction and malabsorption for substantial weight loss, but involve a complex hormonal system that can lead to significant changes in endothelial and cardiovascular health.

## Figures and Tables

**Figure 1 biomedicines-13-00381-f001:**
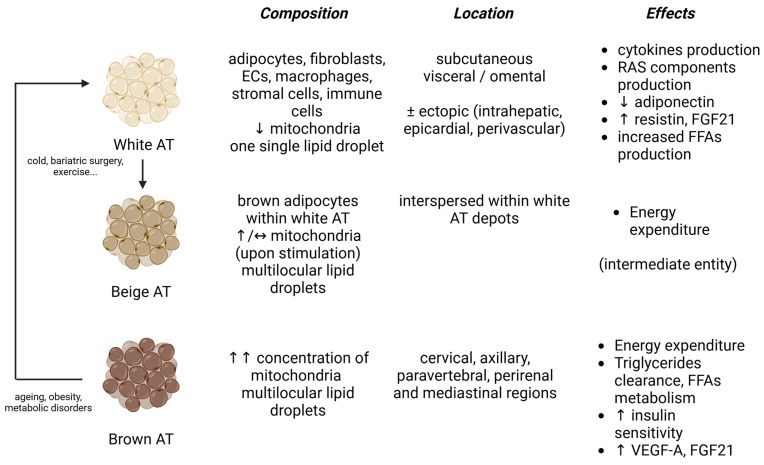
Different types of adipose tissue. White adipose tissue (WAT) is the most represented type of AT, while brown represents around 4% of total fat and beige 1–2%. White and brown AT originate from cells being different both from the structural and functional standpoints. It has a lower concentration of mitochondria compared to other AT types. WAT directly mediates endothelial dysfunction via the secretion of cytokines and mediators involved in vascular pathophysiology. Brown adipose tissue (BAT) bears a thermogenic action and thus it displays higher numbers of mitochondria. It increases triglyceride clearance and insulin sensitivity and produces mediators involved in the angiogenesis and the lowering of the vascular inflammation, globally playing a cardio- and a vasculo-protective function. Beige adipose tissue is an intermediate entity which appears less detrimental to CV health, with research still ongoing. Upon stimulation, WAT can transform into beige (“WAT browning”), with a mechanism that is reversible upon the stimulus’ withdrawal. The continuous arrows between WAT-beige AT and between BAT-WAT indicate the possibility of AT transformation depending on the presence of different stimuli.. Similarly, BAT can undergo a “whitening” process, especially in the presence of metabolic alterations. ↓: decrease; ↑: increase; AT: adipose tissue, ECs: endothelial cells, FGF21: fibroblast growth factor 21, FFAs: free fatty acids. Image created with Biorender (www.biorender.com, accessed on 27 January 2025).

**Figure 2 biomedicines-13-00381-f002:**
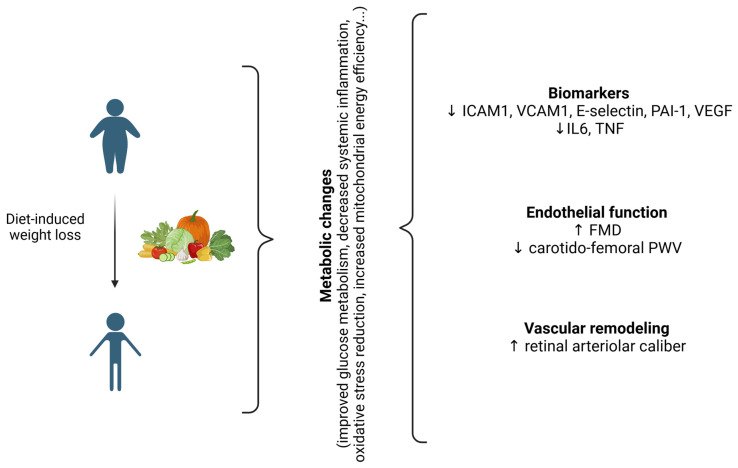
Schematic representation of potential vascular effects of diet-induced weight loss (DIWL). Diet-induced weight loss induces metabolic changes and positively influence endothelial health. After DIWL, a decrease in circulating biomarkers of vascular and endothelial damage is described, together with an improvement in endothelial function and remodelling (increased FMD, decreased pulse wave velocity, increased retinal arteriolar calibre). ↓: decrease; ↑: increase. ICAM-1: intercellular adhesion molecule 1, VCAM-1: vascular cell adhesion molecule 1, PAI-1: plasminogen activator inhibitor 1, VEGF: vascular endothelial growth factor, IL6. Interleukin 6, TNF: tumor necrosis factor, FMD: flow-mediated dilation, PWV: pulse wave velocity. Image created with Biorender (www.biorender.com, accessed on 27 January 2025).

**Figure 3 biomedicines-13-00381-f003:**
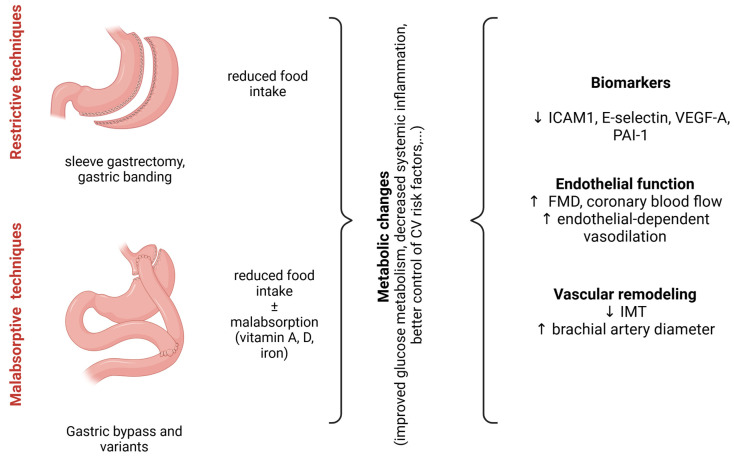
Schematic representation of vascular effects of surgery-induced weight loss (SIWL). After bariatric surgery, an improvement in circulating biomarkers involved in vascular homeostasis is observed. Moreover, patients display improved endothelial function (increased FMD, and endothelium-dependent vasodilation) and vascular remodelling, represented by a decrease in intima-media thickness and an increase in brachial arterial diameter. Notably, hybrid surgical techniques exist combining both restrictive and malabsorptive BS. ↓: decrease; ↑: increase. CV: cardiovascular, ICAM1: intercellular adhesion molecule 1, VEGF-A: vascular endothelial growth factor A, PAI-1: plasminogen activator inhibitor-1, FMD: flow-mediated dilation, IMT: intima-media thickness, RYGB: Roux-en-Y gastric bypass, BPD: biliopancreatic diversion. Image created with Biorender (www.biorender.com, accessed on 27 January 2025).

**Figure 4 biomedicines-13-00381-f004:**
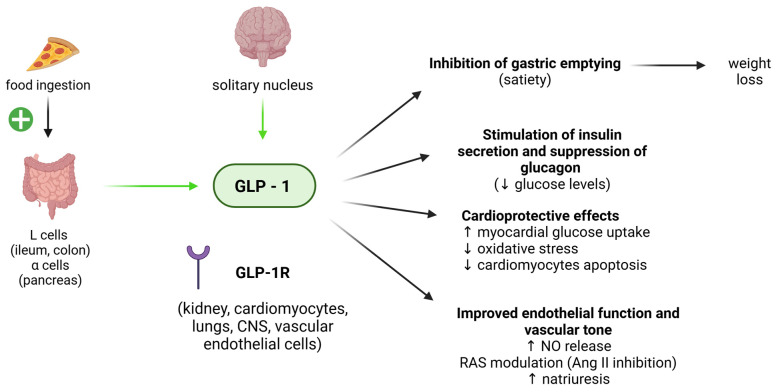
The main actions that GLP-1 exerts on the cardiovascular system, based on the available scientific evidence from human and animal models. GLP-1 secretion is stimulated by food transit in the gastrointestinal tract. GLP-1 exerts metabolic and vasculo-protective actions, both contributing to an improvement in the CV outcome. For instance, GLP-1 can increase myocardial glucose uptake and decrease oxidative stress and cardiomyocytes apoptosis, while improving NO release and natriuresis, positively affecting vascular tone. ↓: decrease; ↑: increase. Positive signs in green indicate stimulation. Green arrows indicate production. GLP-1: glucagon-like peptide 1; GLP-1R: GLP-1 receptor; CNS: central nervous system; NO: nitric oxide; RAS: renin angiotensin system; Ang II: angiotensin II. Image created with Biorender (www.biorender.com, accessed on 27 January 2025).

**Figure 5 biomedicines-13-00381-f005:**
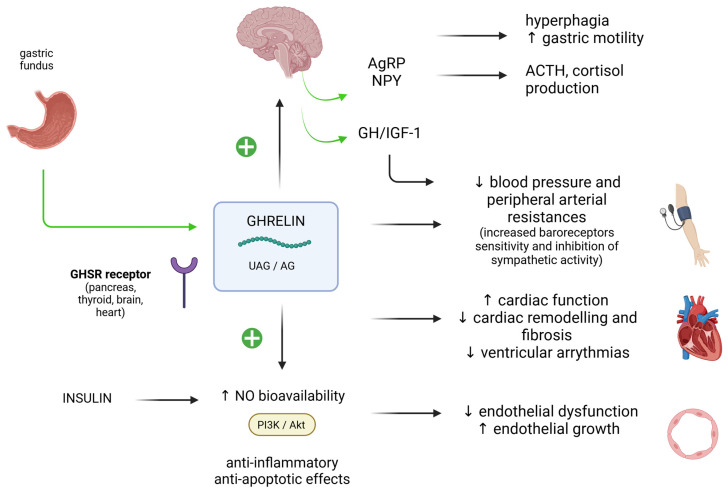
The Main actions of ghrelin on the cardiovascular system, based on available scientific evidence from human and animal models. In the conditions of a negative energy balance, ghrelin is produced by the gastric fundus and induces the synthesis of anabolic mediators at the central nervous system level. In parallel, it can impact the CV system. Preliminary data indicate that unacylated ghrelin (UAG) ameliorates vascular endothelial function, as compared to the acylated form (AG), but further research is needed. A crosstalk between ghrelin action and glucose metabolism is observed, as insulin contributes to increased NO bioavailability via the activation of PI3K/Akt pathway, as does ghrelin. ↓: decrease; ↑: increase. Positive signs in green indicate stimulation. Green arrows indicate production. GHS-R: growth hormone segretagogue receptor, GOAT: ghrelin O-acyltransferase, AgRP: agouti-related protein, NPY: neuropeptide Y, ACTH: adrenocorticotropic hormone, GH: growth hormone, IGF-1: insulin-like growth factor 1, NO: nitric oxide, PI3K: phosphoinositide 3-kinase, Akt: Akt serine-threonine protein kinase. Image created with Biorender (www.biorender.com, accessed on 27 January 2025).

**Figure 6 biomedicines-13-00381-f006:**
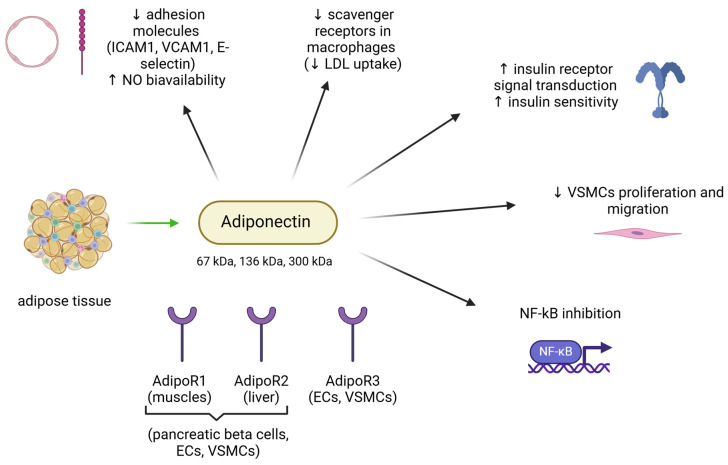
The main actions adiponectin exerts on the cardiovascular system, based on the available scientific evidence from human and animal models. In healthy conditions, adiponectin is produced by the adipocytes in three isoforms with different molecular weights. The beneficial roles of adiponectin on the cardiovascular system are mediated by three peripheral receptors (AdipoR1 to 3) that are expressed in different cell types, including endothelial cells and vascular smooth muscle cells. Adiponectin can improve endothelial function by directing increasing NO bioavailability and reducing LDL uptake and accumulation within the vascular wall. It can also ameliorate glucose metabolism (indirectly promoting vascular homeostasis) and attenuates negative vascular remodelling via decreased vascular smooth muscle cells migration and proliferation. Ultimately, it blocks NF-kB signalling, reducing vascular inflammation and apoptosis. ↓: decrease; ↑: increase. Green arrow indicates production. ICAM-1: intercellular adhesion molecule 1, VCAM-1: vascular cell adhesion molecule 1, NO: nitric oxide, LDL: low-density lipoprotein, VSMCs: vascular smooth muscle cells, NF-kB: nuclear factor kB, ECs: endothelial cells. Image created with Biorender (www.biorender.com, accessed on 27 January 2025).

**Figure 7 biomedicines-13-00381-f007:**
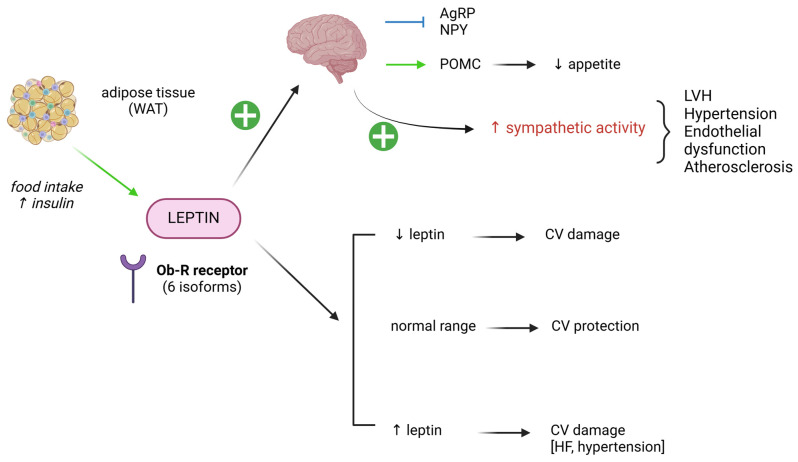
The main actions leptin exerts on the cardiovascular system, based on the available scientific evidence from human and animal models. Leptin is mainly produced by adipose tissue in response to increased food intake and insulin levels. Its action is essentially anorexigenic (stimulates satiety), but different effects on the CV system were described—predominantly in animal models—suggesting that only normal leptin values provide CV protection, whereas any other variation from a set point might negatively affect the CV system. Besides, leptin can stimulate sympathetic activity, an action that is maintained even in leptin-resistance conditions. ↓: decrease; ↑: increase. Positive signs in green indicate stimulation. Green arrows indicate production. Blue line indicates inhibition. WAT: white adipose tissue, AgRP: agouti-related protein, NPY: neuropeptide Y, POMC: proopiomelanocortin, LVH: left ventricular hypertrophy, CV: cardiovascular. Image created with Biorender (www.biorender.com, accessed on 27 January 2025).

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
