# Peer review of "Effects of Weight Loss on Endothelium and Vascular Homeostasis: Impact on Cardiovascular Risk"

_biomedicines, 2025, doi:10.3390/biomedicines13020381_

Round 1
Reviewer 1 Report
Comments and Suggestions for Authors
Thank you for the opportunity to review the manuscript entitled "Effects of weight loss on endothelial function and vascular homeostasis: impact on cardiovascular risk".
I request the authors respond to the following comments:
1) Introduction - lines 45-46: the definition of obesity varies depending on the region of the world (when it comes to BMI), and other elements such as waist circumference or percentage of body fat are also taken into account - I suggest supplementing this element of the manuscript.
2) Introduction - lines 50-52: the authors address the topic of metabolic syndrome without explicitly mentioning it. I recommend incorporating a comprehensive definition of metabolic syndrome together with its separate components. I recommend supplying more comprehensive data regarding the cardiovascular risks linked to obesity, for instance, referencing the studies 10.3390/jcm13071931, 10.1186/s12944-018-0856-8, 10.3390/ijerph18010281, among others.
3) Sections 3 and 4 are de facto subsections of section 2 - I suggest changing them to 2.1 and 2.2 in the revised version of the manuscript.
4) Considering the different metabolic effects of each of the diets described in lines 117-118, it seems unjustified to present the effects on vascular endothelium collectively - I suggest considering presenting this in a broader context, with a brief description of the effects for the Mediterranean diet, VLCD, low-fat, low-carbs, CHO and optionally, others - the following papers may be helpful: 10.3390/ijerph191912762, 10.1001/jama.292.12.1440, 10.1016/j.atherosclerosis.2017.12.012, 10.1007/s11845-022-02944-9, 10.1161/ATVBAHA.120.314838 and others.
5) In line 210 the authors describe the use of Figure 2, but this was used earlier - was there an additional figure omitted?
6) By analogy to the previous ones, sections 6 and 7 are de facto subsections of section 5 - I suggest changing the numbering of the headings.
The work possesses potential but necessitates substantial revisions in alignment with the comments provided.
I will gladly undertake the second review after the revision.
Author Response
Comment 1: “Introduction - lines 45-46: the definition of obesity varies depending on the region of the world (when it comes to BMI), and other elements such as waist circumference or percentage of body fat are also taken into account - I suggest supplementing this element of the manuscript”.
We thank the Reviewer for this comment. We changed the text accordingly, including the different definitions of obesity (see page 2, lines 45-50). We hope that it now provides a more comprehensive definition of the disease, particularly taking into account visceral obesity.
Comment 2: “Introduction - lines 50-52: the authors address the topic of metabolic syndrome without explicitly mentioning it. I recommend incorporating a comprehensive definition of metabolic syndrome together with its separate components. I recommend supplying more comprehensive data regarding the cardiovascular risks linked to obesity, for instance, referencing the studies 10.3390/jcm13071931, 10.1186/s12944-018-0856-8, 10.3390/ijerph18010281, among others”.
We agree with the Reviewer that the text did not specify this issue. We corrected the introduction accordingly (see page 2, lines 56-58).
Comment 3: “Sections 3 and 4 are de facto subsections of section 2 - I suggest changing them to 2.1 and 2.2 in the revised version of the manuscript”.
We agree with the Reviewer. This was changed in the text accordingly (see page 5, line 146; page 6, line 166, page 8, line 234, page 8, line 256).
Comment 4: “Considering the different metabolic effects of each of the diets described in lines 117-118, it seems unjustified to present the effects on vascular endothelium collectively - I suggest considering presenting this in a broader context, with a brief description of the effects for the Mediterranean diet, VLCD, low-fat, low-carbs, CHO and optionally, others - the following papers may be helpful: 10.3390/ijerph191912762, 10.1001/jama.292.12.1440, 10.1016/j.atherosclerosis.2017.12.012, 10.1007/s11845-022-02944-9, 10.1161/ATVBAHA.120.314838 and others”.
We thank the Reviewer for this comment and for providing some interesting references on this subject. We’ve corrected the text accordingly and addressed this issue in the final section of the paper as well (see page 5, lines 129-132; page 18, line 562-563).
Comment 5: “In line 210 the authors describe the use of Figure 2, but this was used earlier - was there an additional figure omitted?”
We thank the Reviewer for pointing out this mistake. This was corrected (see page 8, line 227).
Comment 6: “By analogy to the previous ones, sections 6 and 7 are de facto subsections of section 5 - I suggest changing the numbering of the headings”.
This was corrected as above (please see page references in Comment 3).
Reviewer 2 Report
Comments and Suggestions for Authors
-
The title “Effects of weight loss on endothelial function and vascular homeostasis: implications for cardiovascular risk” is not consistent with the main content of the manuscript, “How methods of weight loss dieting and surgery affect endothelial function and cardiovascular risk.” Since, the author already points out that a lot of studies indicate that studies on DIWL showed that FMD improvements may not correlate with weight loss.
-
Figure 1 There are no data “showing the different types of adipose tissue and their relevance to cardiovascular health”. There are only data showing that WAT adipose tissue directly mediates endothelial dysfunction through the secretion of cytokines and mediators involved in vascular pathophysiology, what about the other two types of adipose tissue like Brown AT and Beige AT?
-
What's CRP? Please provide the full name/description of the abbreviation when it first appears.
-
The authors have pointed out that many studies on DIWL suggest that improvements in FMD may not be associated with weight loss. For this reason, it is best to avoid the use of solid arrows in Figure 2.
-
The author did not provide ENDOTHELIAL FUNCTION information in “5. SURGERY-INDUCED WEIGHT LOSS AND ENDOTHELIAL FUNCTION”. Only descript several types of BS.
-
Between the statements “(GLP-1) is a hormone secreted by the gastrointestinal tract and central nervous system (CNS) in response to ingested food” and “levels of GLP-1 are lower in people with obesity, especially in the post-prandial period” there is a slight gap in understanding. The more you eat also means the more food ingestion and the more you eat the more likely you are to be obese. But why GLP- is lower in people with obesity? The authors need further clarification here.
-
In the manuscript, the authors state that “acylated gastrin improves parameters of cardiac function but not endothelial function, and administration of unacylated gastrin to patients with metabolic syndrome significantly improves endothelium-dependent vasodilatation and increases nitrogen oxide availability.” However, in the legend to Figure 4, the authors write that it is unclear whether the cardiovascular and metabolic effects are mediated by unacylated gastrin (UAG) or acylated gastrin (AG), or which isomer is more beneficial to the cardiovascular system. In Figure 4, the authors again use solid arrows to indicate that UAG to AG is more beneficial for cardiovascular and endothelial function. All of this is too confusing. The authors need to give a clear or rational explanation here.
Author Response
“Comment 1: The title “Effects of weight loss on endothelial function and vascular homeostasis: implications for cardiovascular risk” is not consistent with the main content of the manuscript, “How methods of weight loss dieting and surgery affect endothelial function and cardiovascular risk.” Since, the author already points out that a lot of studies indicate that studies on DIWL showed that FMD improvements may not correlate with weight loss”.
We thank the Reviewer for this advice. We then changed the title into “Effects of weight loss on endothelium and vascular homeostasis: impact on cardiovascular risk” (see page 1, lines 2-3).
Comment 2: “Figure 1 There are no data “showing the different types of adipose tissue and their relevance to cardiovascular health”. There are only data showing that WAT adipose tissue directly mediates endothelial dysfunction through the secretion of cytokines and mediators involved in vascular pathophysiology, what about the other two types of adipose tissue like Brown AT and Beige AT?”
We agree with the Reviewer on this comment, that helped us to clarify the text. Indeed, the data presented in the Figure are not all related to CV health. This is mainly because WAT is the most studied when it comes to endothelial dysfunction and CV risk, while the action of the other types of adipose tissue (especially beige AT) on endothelial cells and the CV system are still a matter of research. In our corrections, we modified the Figure legend to underline this point and slightly modified the Figure 2 according to the most recent evidence. We hope these corrections will meet your requirements (see page 4, lines 104-109 and page 6, lines 111-117).
Comment 3: “What's CRP? Please provide the full name/description of the abbreviation when it first appears”.
We thank the Reviewer for pointing out this mistake. This was corrected (see page 6, line 186).
Comment 4: “The authors have pointed out that many studies on DIWL suggest that improvements in FMD may not be associated with weight loss. For this reason, it is best to avoid the use of solid arrows in Figure 2”.
We thank the Reviewer for this correction. Indeed, we changed both Figure 2 and Figure 3 to underline this point (see page 7, line 208 and page 10, line 280).
Comment 5: “The author did not provide ENDOTHELIAL FUNCTION information in “5. SURGERY-INDUCED WEIGHT LOSS AND ENDOTHELIAL FUNCTION”. Only descript several types of BS”.
We thank the Reviewer for this comment. Based on the suggestion of the other Reviewer, we changed the structure of the text making sub-headings of section 2 and 3 (see page 5, line 146, page 6, line 166, page 8, line 234 and page 8 line 256). Thus, the section 3 now includes both endothelial function and circulating biomarkers data.
Comment 6: “Between the statements “(GLP-1) is a hormone secreted by the gastrointestinal tract and central nervous system (CNS) in response to ingested food” and “levels of GLP-1 are lower in people with obesity, especially in the post-prandial period” there is a slight gap in understanding. The more you eat also means the more food ingestion and the more you eat the more likely you are to be obese. But why GLP- is lower in people with obesity? The authors need further clarification here”.
We thank the Reviewer for pointing it out. In people who are not overweight, GLP-1 is triggered by eating to start the "satiety" response, a process that stops excessive food consumption and avoids weight gain. In individuals with obesity, as well as in those with pre-diabetes, this mechanism is impaired, resulting in patients' inability to accurately halt eating. The reasons behind the insufficient rise in GLP-1 levels following meals are unknown, and scientists are debating whether this phenomenon occurs prior to or subsequent to weight gain. We then modified the text following your comment. We hope that it now conveys a clearer message (see page 11, lines 314-316).
Comment 7: “In the manuscript, the authors state that “acylated gastrin improves parameters of cardiac function but not endothelial function, and administration of unacylated gastrin to patients with metabolic syndrome significantly improves endothelium-dependent vasodilatation and increases nitrogen oxide availability.” However, in the legend to Figure 4, the authors write that it is unclear whether the cardiovascular and metabolic effects are mediated by unacylated gastrin (UAG) or acylated gastrin (AG), or which isomer is more beneficial to the cardiovascular system. In Figure 4, the authors again use solid arrows to indicate that UAG to AG is more beneficial for cardiovascular and endothelial function. All of this is too confusing. The authors need to give a clear or rational explanation here”.
We agree with the Reviewer on the need for having a better comprehension of the paragraph and the associated Figure. Generally speaking, available scientific evidence indicates that UAG exerts a positive impact on the endothelium, both directly (improved endothelial function, increased NO production) and indirectly (improved glucose metabolism), while evidence concerning AG is not conclusive, possibly indicating a cardiac tropism rather than an effect on the vasculature. Moreover, several studies do not indicate the specific isoform studied, thus limiting our analysis. We tried to underline these differences correcting the main text and changing Figure 4 and the Figure legend accordingly. We do hope these modifications will improve the quality and comprehension of this section (see page 13, lines 370-372 and 382-385, page 14, lines 410-413, Figure 4).
Round 2
Reviewer 1 Report
Comments and Suggestions for Authors
The authors have responded appropriately to my previous comments. I have no additional comments.
Reviewer 2 Report
Comments and Suggestions for Authors
The manuscript is a significant improvement over the previous manuscript in terms of writing and data arrangement. The authors responded appropriately to most of the issues raised by the reviewers.
Comments on the Quality of English LanguageNA